

# Taxonomic, biological and geographical traits of species in a coastal dune flora in the southeastern Cape Floristic Region: regional and global comparisons

Richard M. Cowling[1,2], Caryl Logie[3], Joan Brady[2], Margie Middleton[3] and B. Adriaan Grobler[1]

[1] African Centre for Coastal Palaeoscience, Botany Department, Nelson Mandela University, Port Elizabeth, Eastern Cape, South Africa
[2] Friends of St Francis Nature Areas (FOSTER), Cape St Francis, Eastern Cape, South Africa
[3] Custodians of Rare and Endangered Wildflowers (CREW): Fourcade Botanical Group, St Francis Bay, Eastern Cape, South Africa

Corresponding author
Richard M. Cowling,
rmc@kingsley.co.za

## ABSTRACT

In Mediterranean-Climate Ecosystems (MCEs), Holocene coastal dunes comprise small, fragmented and dynamic features which have nutritionally imbalanced and excessively drained, droughty, sandy soils. These characteristics, along with summer drought and salt-laden winds, pose many challenges for plant colonization and persistence. Consequently, MCE dune floras are likely to be distinctive with a high proportion of habitat specialists and strong convergence in growth form mixes. Very little research has compared the species traits of dune floras within and across MCEs. This paper contributes to filling that gap. Here, we analyze the taxonomic, biological and geographical traits for all 402 species in a flora from a dune landscape (Cape St Francis) in the southeastern Cape Floristic Region (CFR) and compare patterns with the trait profiles of other dune floras at a regional (CFR) and global (MCE) scale. Within the CFR, the southeastern (all-year-rainfall) flora at Cape St Francis had a similar trait profile to western (winter-rainfall) dune floras, except for having a lower representation of species belonging to CFR-endemic clades, and higher number of species associated with tropical lineages. The St Francis flora, in common with other CFR and MCE floras, was dominated by members of the Asteraceae, Fabaceae and Poaceae. Some 40% of the St Francis flora was endemic to the CFR, typical of the high rate of MCE-level endemism elsewhere in the CFR, and in other MCEs. About 30% of the flora was confined to calcareous sand, a value typical for many other MCE sites. The St Francis flora, as well as other CFR dune floras, differs from those of other MCEs by having many species associated with shrubby lineages, and by the relatively high incidence of species associated with tropical lineages. The growth form profile of the St Francis and other CFR floras shows strongest similarity with that of Australian MCE dunes in that in both regions, evergreen hemicryptophytes and shrubs share dominance, and annuals are floristically and ecologically subordinate. The least similar of MCEs to the St Francis trait profile is the Mediterranean Basin where annuals are the most frequent growth form while shrubs are subordinate. California and Chile dune floras appear to occupy an intermediate position, in terms of growth form mix, between the Cape and Australia on the one hand, where dune floras have retained features typical of

nutrient-poor soils, and the Mediterranean Basin, where dwarf, deciduous shrubs and annuals dominate the life form spectrum. All MCE dunes are threatened by alien plants, infrastructure development, tourism demands and rising sea levels. The high incidence of species of conservation concern in CFR dune floras underestimates the exponentially increasing threats to their habitats, which are already historically at a much-reduced extent. All remaining coastal dune habitat in the CFR, and probably in other MCEs, should be conserved in their entirety.

## INTRODUCTION

Landscapes comprised of young, calcareous, coastal dunes pose several challenges for plant colonization and persistence (*Maun, 2009*). Owing to Pleistocene sea-level fluctuations, these landscapes are repeatedly eroded and rebuilt elsewhere in different configurations, depending on the topography of the coastline (*Barbour, De Jong & Pavlik, 1985*; *McLachlan & Brown, 2006*). They are often subjected to incursions of younger, mobile sand plumes which transgress stable habitats (*Illenberger & Burkinshaw, 2008*; *Bateman et al., 2011*). Soils are sandy, excessively drained and highly alkaline, resulting in low water holding capacity and low fertility, the latter a consequence of high pH restricting the uptake of key nutrients (*Brady, 1974*). Furthermore, coastal dunes, especially on their seawards margin, are subject to frequent, strong, salt-laden winds (*Wilson & Sykes, 1999*; *McLachlan & Brown, 2006*). Plants growing on coastal dunes need to have overcome considerable ecophysiological constraints. Therefore, dune floras are likely to be floristically and functionally distinctive relative to adjacent inland floras, be poor in species, and have a high proportion of dune specialists (edaphic endemics) (*Cowling, Holmes & Rebelo, 1992*; *Van Der Maarel & Van Der Maarel-Versluys, 1996*; *Brunbjerg et al., 2014*).

Coastal dunes are present everywhere except Antarctica, although they are best developed in sub-humid, warm temperate to subtropical climates (*Doing, 1985*; *Hesp, 2007*). They are particularly well developed in Mediterranean-Climate Ecosystems (MCEs) (*Peinado et al., 2007*), including the Cape Floristic Region (CFR) (*Tinley, 1985*). Their extent at any given time is a function of sand supply (mostly from rivers), local topography, and rainfall and wind regime. In semi-arid areas with high sand supply, strong winds and low relief, sand plumes may extend for several kilometers inland; along steeply falling coasts with low sand supply and light winds, dunes may occupy a narrow cordon only tens of meters wide (*Tinley, 1985*; *Roberts et al., 2006*).

Most coastal dunes in the CFR are relatively young, having been deposited with rising sea levels since the start of the Holocene (*Roberts et al., 2006*; *Bateman et al., 2011*). Our focus is on these young dunes and not the older Quaternary and Neogene dunes where weathering has produced oxidized, leached and lithified (calcarenite) soils (*Tinley, 1985*), which support different vegetation types and associated floras

(*Cowling & Holmes, 1992*). Holocene dunes today occupy a small part of the Cape coast with the largest exposures associated with Table Bay, Walker Bay, Wilderness–Sedgefield, St Francis Bay and Algoa Bay (*Tinley, 1985*; *Roberts et al., 2006*).

There is disagreement in the literature whether Cape dune vegetation should be attributed to the Subtropical Thicket biome (*Low & Rebelo, 1996*; *Cowling & Heijnis, 2001*) or the Fynbos biome (*Mucina & Rutherford, 2006*). The reason for this is that the vegetation comprises a matrix of fynbos and subtropical thicket, the former prevailing in fire-exposed and edaphically dry sites, the latter dominating in moist and fire-protected sites where it can attain forest stature (*Lubke, 1983*; *Cowling, 1984*; *Tinley, 1985*; *Avis & Lubke, 1996*; *Cowling et al., 1997*). Biomass of dune fynbos is dominated by species endemic to Holocene dunes although non-endemics comprise most of the species growing there (*Cowling, 1983*). Dune thicket, comprising lineages of tropical origin, is a distinctive formation that includes several species endemic to the coastal dunes of the CFR (*Cowling, 1983*; *Vlok, Euston-Brown & Cowling, 2003*).

Most botanical research on Cape dunes has focused on the strand plant hummock dunes that front the beach (*Boucher & Le Roux, 1993*; *Taylor & Boucher, 1993*; *Lubke et al., 1997*). There has been much less research on the flora and vegetation of vegetated back dunes. Little is known about the trait composition of the dune floras of the CFR and how this compares with dune floras of other MCEs. Here, we report on the taxonomic, biological and geographic traits of the species of a Holocene dune flora from Cape St Francis in the southeastern part of the CFR. Biological traits comprised woody and herbaceous growth forms, woody species post-fire regeneration, and incidence of succulence; geographic traits comprised biome membership, phytogeographical affinity and level of endemism (spatially and edaphically). We also assessed the conservation status of all species in the flora. We discuss the results for the St Francis flora (all-year rainfall) in the context of trait patterns of floras from the winter-rainfall part of the Cape. In order to provide a global perspective, we compare trait patterns for Cape dune floras with those from the world's four other MCEs.

## STUDY AREA

The study area consists of seven protected areas covering a total of 813 ha in the Cape St Francis region of the southeastern CFR (Fig. 1). The landscape is underlain by Paleozoic rocks of the Cape Supergroup, notably quartzitic sandstone of the Peninsula Group (that form the erosion resistant points and rocky shores) and the softer sandstones of the Goudini Formation (which underlies the sandy embayments and their hinterlands) (*Le Roux, 2000*). The overlying sediments comprise Holocene aeolianites (Schelmhoek Formation) deposited on Pleistocene calcarenites (Nahoon Formation). Contrary to *Le Roux (2000)*, who undertook a geological survey (1:250,000 scale) of a region that included the study area, we could identify no Cenozoic aeolianites (Nanaga Formation) there.

Geomorphologically, the study area comprises vegetated parabolic dunes and several more recent plumes of mobile sand forming an impressive (maximum 14 km-long) bypass dune system (*Tinley, 1985*; *Illenberger & Burkinshaw, 2008*). On the inland margin,

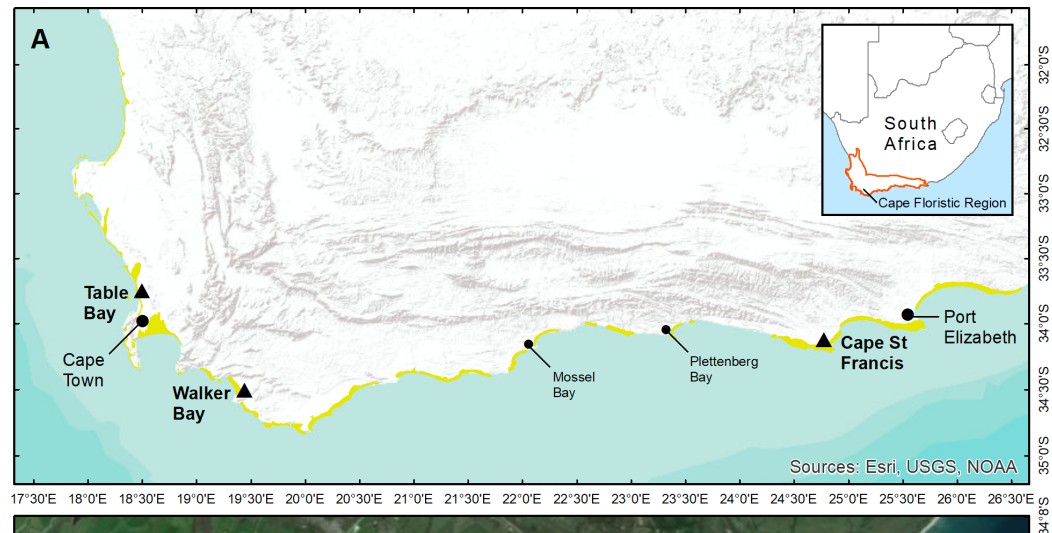

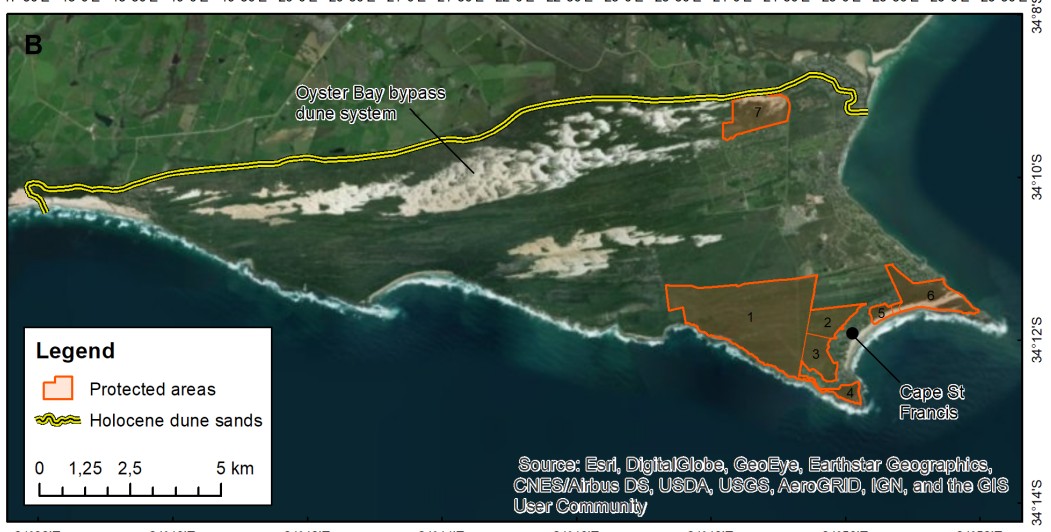

**Figure 1 The area investigated during this study.** (A) Distribution of Holocene coastal dunes (yellow) in the Cape Floristic Region (CFR) of South Africa. The study area, Cape St Francis, and other CFR dune sites used for comparisons are indicated with triangles. Imagery sources: Esri, USGS, NOAA. (B) The Holocene dune landscape around Cape St Francis in the southeastern CFR. The yellow line indicates the boundary between calcareous Holocene aeolianites (also including minor Pleistocene calcarenite outcrops) and other substrata. Protected areas sampled for flora composition are: 1—Rocky Coast Farm; 2—Air Park Reserve; 3—Irma Booysen Nature Reserve; 4—Seal Point Nature Reserve; 5—Seal Bay Nature Reserve; 6—Cape St Francis Nature Reserve; 7—St Francis Links Nature Reserve. Imagery sources: Esri, DigitalGlobe, GeoEye, Earthstar Geographics, CNES/Airbus DS, USDA, USGS, AeroGRID, IGN.

dunes reach a maximum altitude of 62 m. The bypass system is now largely dysfunctional with large tracts of sand stabilized by vegetation at the source, and urban development at the sink (Fig. 1). Most of the study area is associated with vegetated dunes although it does include parts of the Oyster Bay bypass dune in the St Francis Links nature reserve, and the small but intact bypass dune in the Cape St Francis nature reserve. Other physical features of the study area include strand plant hummock dunes, wind- and salt-exposed rocky shores, and three wetland types.

Soils of the vegetated dunes are deep (>1 m), well-drained apedal sands (Fernwood form) of moderate alkalinity, with pH values between 7.0 and 8.0, high exchangeable cation status and relatively high available phosphorus (20–50 ppm) (*Cowling, 1984*). Thicket topsoils have a slightly higher organic carbon and nitrogen content than fynbos soils (*Cowling, 1984*, *1990*), probably because of the low incidence of fire in thicket and its effects as a mineralizing agent (*Stock & Allsopp, 1992*). Mobile sands have higher alkalinity than vegetated dunes, and no measureable organic matter. Wetland soils are variously hydromorphic and organic rich.

The climate of the region is warm-temperate with a sub-Mediterranean rainfall regime (*Cowling, 1984*). The mean annual temperature at Cape St Francis (seven m) is 17.1 °C with absolute minima of 4 °C and maxima in the lower thirties, usually associated with berg winds in autumn and early winter. Mean annual rainfall is 673 mm. While high rainfall may occur in any month of the year, the three warmest months (December–February) consistently record the lowest rainfall. Plant growth and reproduction is concentrated in the cooler months, except for most thicket species which, in keeping with their tropical origins, have a largely warm-season phenology (*Pierce & Cowling, 1984*). The wind regime at Cape St Francis is fierce with strong winds from the west-southwest in winter and from east-southeast in summer. The windiest period is spring and early summer, when gales are frequent and calms are rare. The least windy time is late autumn and early winter.

The vegetation of the stable dunes predominantly occurs as a mosaic of two biomes, namely Fynbos and Subtropical Thicket (*Cowling, 1984*). The Fynbos component forms a distinctive assemblage of species dominated by those endemic to calcareous Holocene aeolianites of the Cape south coast (e.g., *Metalasia muricata*, *Restio eleocharis*, *Carpobrotus deliciosus*, *Erica chloroloma*, *Salvia africana-lutea*) (*Cowling, 1983*). Dune thicket is dominated by species of tropical origin and is differentiated from other thicket types by the presence of several species endemic to calcareous dunes, for example, *Olea exasperata*, *Rapanea gilliana*, *Searsia crenata*, *Maytenus procumbens*, *Robsonodendron maritimum* and *Cussonia thyrsiflora*. Other biomes represented in the study area include Forest, Grassland, Coastal and Wetland; together they cover less than 5% of the region. Forest is associated with fire-protected sites such as steep-walled and narrow interdune valleys; characteristic species are *Chionanthus foveolatus*, *Zanthoxylum capense*, *Dovyalis rhamnoides* and *Clausena anisata* (*Cowling et al., 1997*). Grassland is restricted to broad dune valleys where near-surface calcarenite or Table Mountain Group bedrock impedes drainage; typical species are *Themeda triandra*, *Cymbopogon marginatus*, *Helichrysum herbaceum* and *Geranium incanum*. The Coastal biome includes the semi-mobile hummock dunes (*Arctotheca populifolia*, *Scaevola plumieri*, *Silene crassifolia* and *Phylohydrax carnosa*); mobile bypass dunes (*Capeochloa cincta*, *Morella cordifolia*, *Psoralea repens*, *Seriphium* sp. nov.); and the rocky coastline (*Syncarpha sordescens*, *Osteospermum fruticosum*, *Delosperma saxicola*, *Drosanthemum intermedium* and *Sprorobolus virginicus*) (*Taylor & Boucher, 1993*). The Wetland biome encompasses three wetland types, namely coastal seep (*Samolus porosus*, *Senecio halimifolius*, *Helichrysum gymnocomum*, *Chironia peduncularis*), valley bottom (*Cyperus thunbergii*, *Psoralea*

*floccosa, Nidorella pinnatifida, Mentha aquatica, Grammatotheca bergiana*) and dune slacks (that develop in the lee of mobile dunes) (*Juncus krausii, Ficinia nodosa, Centella asiatica, Plecostachys serpyllifolia*) (*Day, 2011*).

Other than the tiny forest patches, frontal dunes and mobile bypass dunes, all biomes are subject to recurrent fire at intervals of 10–15 years (*Cowling et al., 1997*). Well established thicket clumps burn less frequently than the fynbos matrix although most thicket patches over much of the study area were burnt in an extremely hot fire in early January 2016.

## METHODS

### Flora compilation

The flora (excluding alien species) was compiled from several sources, namely (i) collections made by RMC in 1979–1980, (ii) collections made by the Fourcade Botanical Group, a branch of the Custodians of Rare and Endangered Wildflowers, since 2000 (selected specimens from both of these collections are housed in the Selmar Schonland Herbarium at Rhodes University (GRA), (iii) and photographic observations by all contributors over a period of 20 years. Collection of plant specimens was approved by the Provincial Administration of the Eastern Cape Province: Chief Directorate Environmental Affairs (permit to pluck flora No. CRO 138/18CR). Photographs were identified by the authors and their colleagues and these records (630 in total) were collated in an online biodiversity database, iNaturalist (https://www.inaturalist.org/observations? q=StFrancisDuneFlora&search_on=tags), where they were scrutinized by botanical colleagues for several months. We recorded 424 responses and four name changes were made. Nomenclature follows South African National Biodiversity Institute (http://posa.sanbi.org).

To facilitate comparisons with other dune floras in the CFR, we extracted floras for Grootbos Private Nature Reserve (Walker Bay) (*Privett & Lutzeyer, 2010*) on the eastern edge of the winter-rainfall region, and for Koeberg (Table Bay) (*Low, 2011*) in the strongly winter-rainfall region, and categorized component species in terms of all the traits described below for Table Bay and only floristic traits and conservation status for the Walker Bay flora. These data are available as Datas S2 and S3, respectively.

### Geographical traits

Using *Cowling (1984)* and our own field experience, we allocated each species to one of the following biomes: Fynbos, Subtropical Thicket (hereafter Thicket), Forest, Grassland, Coastal, Wetland and Disturbed (Fig. 2). In the case of species that occurred in more than one biome (e.g., *Pterocelastrus tricuspidatus*, which grows in Thicket and Forest), we allocated them to the biome in which they were most abundant (in this case Thicket).

Based on information in *Goldblatt (1978)*, *Born, Linder & Desmet (2007)*, *Manning & Goldblatt (2012)*, as well as several reliable internet sources, we categorized each species based on the distribution of its respective genus in relation to the following phytogeographic regions: Greater Cape Floristic Region (GCFR), Southern Africa (SA),

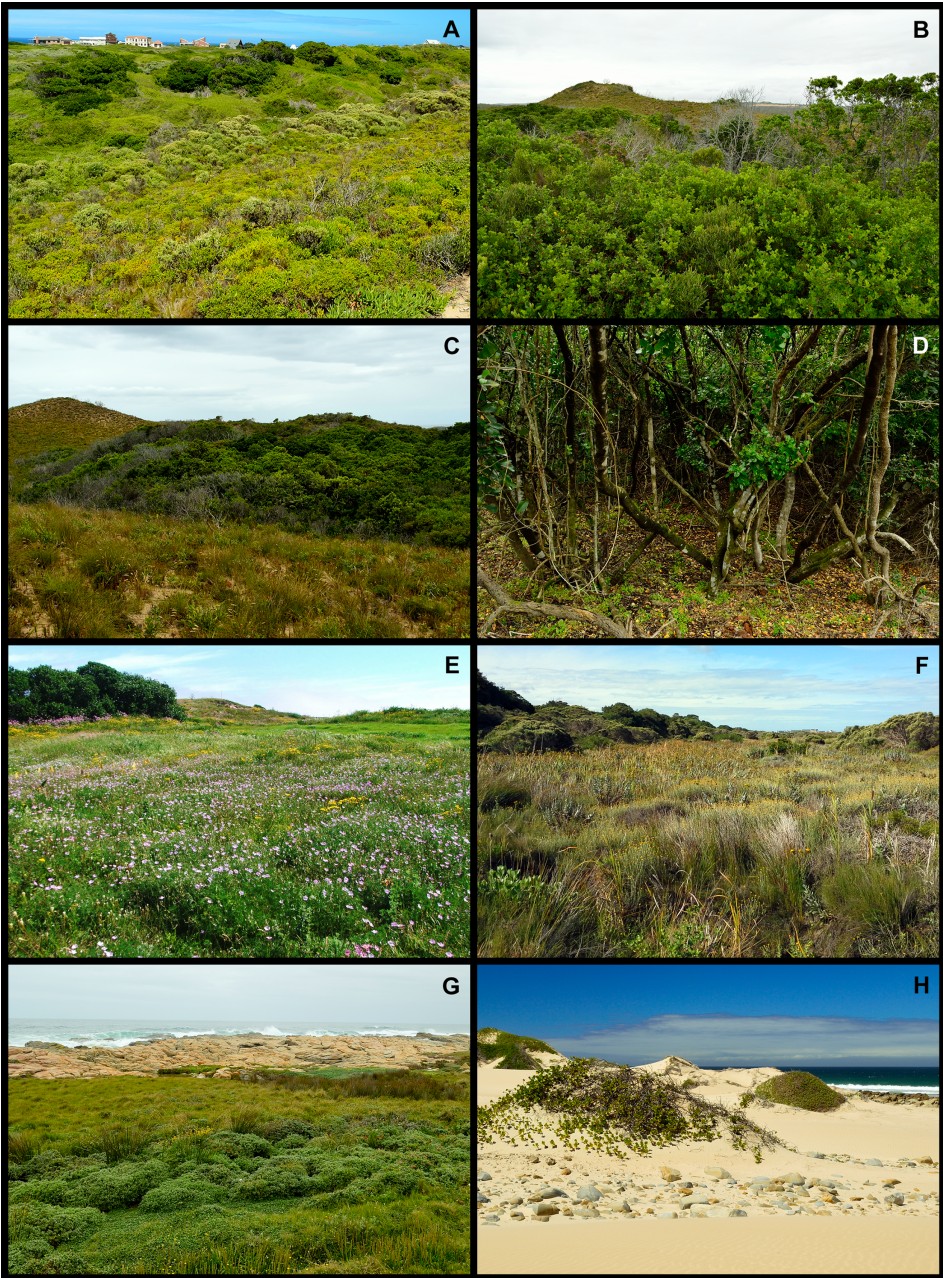

**Figure 2 Examples of the biomes that occur in the Holocene dune landscape around Cape St Francis in the southeastern Cape Floristic Region.** (A) Fynbos dominated by the low shrubs *Agathosma apiculata* and *Metalasia muricata*. In the background, a patch of Subtropical Thicket dominated by *Pterocelastrus tricuspidatus*, is visible. (B) Subtropical Thicket dominated by *Olea exasperata* and *Searsia glauca*. (C) A pocket of Forest occurring in a fire-protected interdune valley, with recently burnt fynbos (ca 1 year old) on surrounding dune slopes and crests. (D) Forest shares many species with Subtropical Thicket, but it is structurally different from the latter in having distinct strata with the canopy and ground layer well separated. (E) Grassland, comprising mostly grass and forb species, is here dominated by *Geranium incanum*. (F) One example of the Wetland biome with various sedges (Cyperaceae), *Phragmites australis* and *Typha capensis* prominent. (G) The Coastal biome includes rocky shorelines where the low shrub *Syncarpha sordescens* forms extensive stands. (H) Semi-mobile hummock dunes, where *Scaevola plumieri* is common, also form part of the Coastal biome. Photos: B. Adriaan Grobler (A–D, G, H); Caryl Logie (E, F).

Temperate Africa (TeAfr), Tropical Africa (TrAfr), Pantropical (PanTr), Pantemperate (PanTe), Cosmopolitan (Cosmo) (*Cox, 2001*; *Galley et al., 2006*; *Gehrke & Linder, 2009*).

We also categorized species according to their distributions within the GCFR, identifying those endemic to: the entire region (GCFR), the CFR, the southeastern centre of the CFR (*Cowling, 1983*; *Manning & Goldblatt, 2012*; *Colville et al., 2014*); and those with distributions extending beyond the GCFR.

We also categorized species in terms of their edaphic niche, recognizing species with distributions largely restricted to calcareous dune sands and calcarenites, and those also occurring on acid sands and neutral loams of the interior. Data were derived from *Cowling (1983)*, *Cowling & Holmes (1992)* and *Manning & Goldblatt (2012)*.

## Biological traits

We classified species into five woody and five herbaceous growth forms. Decisions were based on the most commonly occurring form of a species throughout the study area and not on information derived from the literature. Thus, high-plasticity species such as *Sideroxylon inerme*, which can grow as a dwarf, multi-stemmed shrub or a tall, single-stemmed tree, occurs most commonly in our area as a multi-stemmed, tall shrub, and was classified as such. Although most fynbos shrubs have limited growth form plasticity, the extreme wind regime of the study area results in generally lower stature than in less wind-exposed coastal dunes of the Garden Route (e.g., Wilderness dunes). Thus, many species classified as low shrubs there would be categorized as dwarf shrubs in the study area. Woody growth forms comprised trees (>5 m, mostly single-stemmed), tall (two–five m) shrubs, low (0.5–1 m) shrubs, dwarf (<0.5 m) shrubs and lianas. Herbaceous growth forms were evergreen hemicryptophytes (mainly Cyperaceae and Restionaceae), deciduous hemicryptophytes (mainly Poaceae), geophytes, annuals, forbs and vines. We also identified species showing succulence, with (putative) CAM physiology (*Mooney, Troughton & Berry, 1977*; *Rundel, Esler & Cowling, 1999*), a trait that transcended other growth forms (but most being dwarf shrubs).

We categorized woody species in terms of their mode of post-fire regeneration, based on the literature (*Cowling & Pierce, 1988*) but mainly on our own observations after a wildfire burnt part of the study area in February 2016. Following *Pausas et al.'s (2004)* schema, species were classed as obligate resprouters (only resprout post fire, seedlings appear in specific microsites between fires), facultative resprouters (both resprout and produce seedlings post-fire) and non-sprouters (adults killed by fire and regenerate from seeds only).

## Species conservation status

Following the online Red List of South African Plants (*SANBI (South African National Biodiversity Institute), 2017*), we categorized species according to their national conservation status. For species that were described after the Red List was compiled, we referred to the relevant taxonomic literature to establish their conservation status (*Bello et al., 2017*). Species that remain undescribed (three species in our flora) were assigned a conservation status of "Data Deficient," while those that could only be

**Table 1** **Largest families (>10 spp.) and genera (>5 spp.) in three coastal dune floras of the Cape Floristic Region.** Data for Table Bay (*Low, 2011*) and Walker Bay (*Privett & Lutzeyer, 2010*) are available as Datas S2 and S3, respectively.

| Cape St Francis (402 spp.) | | Walker Bay (Grootbos) (298 spp.) | | Table Bay (Koeberg) (295 spp.) | |
|---|---|---|---|---|---|
| **Families** | **Genera** | **Families** | **Genera** | **Families** | **Genera** |
| Asteraceae (51) | *Senecio* (10) | Asteraceae (48) | *Ficinia* (10) | Asteraceae (47) | *Ficinia* (9) |
| Poaceae (30) | *Ficinia* (8) | Iridaceae (28) | *Helichrysum* (9) | Aizoaceae (19) | *Crassula* (8) |
| Fabaceae (29) | *Helichrysum* (7) | Scrophulariaceae (18) | *Senecio* (9) | Cyperaceae (18) | *Helichrysum* (7) |
| Cyperaceae (25) | *Indigofera* (6) | Cyperaceae (14) | *Moraea* (8) | Scrophulariaceae (17) | *Ehrharta* (6) |
| Aizoaceae (15) | *Thesium* (6) | Fabaceae (13) | *Aspalathus* (7) | Poaceae (15) | *Oxalis* (6) |
| Iridaceae (13) | *Asparagus* (5) | Poaceae (12) | *Pelargonium* (7) | Iridaceae (12) | *Senecio* (6) |
| Scrophulariaceae (12) | | Aizoaceae (11) | *Oxalis* (6) | Crassulaceae (10) | *Trachyandra* (6) |
| | | | *Searsia* (6) | | *Asparagus* (5) |
| | | | *Cliffortia* (5) | | *Isolepis* (5) |
| | | | *Romulea* (5) | | *Limonium* (5) |
| | | | | | *Pelargonium* (5) |
| | | | | | *Thesium* (5) |

identified to generic level (one species in our flora) were omitted from this categorization. In this paper, we use the collective term "threatened species" to refer to those species that are categorized as Vulnerable, Endangered and Critically Endangered, while "species of conservation concern" (SCCs) include threatened species as well as those categorized as Data Deficient, Rare and Near Threatened. Even though some species with decreasing population trends (*SANBI (South African National Biodiversity Institute), 2017*) are categorized as Least Concern in the Red List, we categorized them as Declining and include them as SCCs.

# RESULTS

## Flora composition

We identified 402 species, 268 genera and 78 families in the 813 ha of nature reserve in the Cape St Francis region. The full list of species, together with their traits, is presented in Data S1. The most speciose families were Asteraceae, Poaceae and Fabaceae (Table 1), which together comprised 27% of the total tally. The three largest genera were *Senecio*, *Ficinia* and *Helichrysum*.

## Geographical traits

Some 40% of the St Francis flora was associated with the Fynbos biome (Fig. 3A), represented largely by Asteraceae (*Helichrysum*, *Senecio*), Fabaceae (*Indigofera*, *Aspalathus*), Poaceae (*Pentameris*, *Tribolium*) and Scrophulariaceae. This result is expected, given that much of the study area supports species-rich fynbos vegetation. The Thicket biome was home to 16.2% of species, best represented by members of the Celastraceae, Anacardiaceae (*Searsia*) and Asparagaceae. Cyperaceae and Asteraceae dominated the Wetland biome flora, which comprised 13.4% of the total, a high tally given the small area this biome occupies in

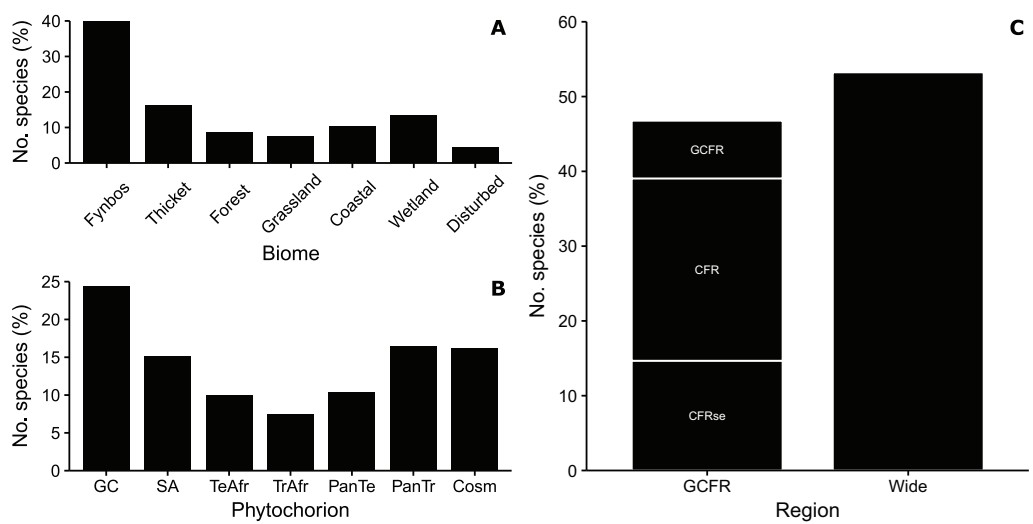

**Figure 3 Geographical traits of a dune flora from Cape St Francis in the southeastern Cape Floristic Region.** (A) The proportion of species associated with each of the biomes occurring in the study area. (B) The proportion of species associated with each phytogeographic region; GC, Greater Cape; SA, Southern Africa; TeAfr, Temperate Africa; TrAfr, Tropical Africa; PanTr, Pantropical; PanTe, Pantemperate; Cosm, Cosmopolitan. (C) The proportion of species endemic to different regions in the Greater Cape Floristic Region; GCFR, Greater Cape Floristic Region; CFR, Cape Floristic Region; CFRse, southeastern centre of endemism in the CFR; Wide, widespread species not endemic to the GCFR. Note that proportions of species in GCFR regions are cumulative (i.e., total CFR endemics = CFRse + CFR, total GCFR endemics = CFRse + CFR + GCFR).      

the study area. Similarly, the Coastal biome—a narrow sliver where plants must be capable of enduring frequent strong, salt-laden winds—supported a rich flora of 41 species (10.2%) comprising mainly Asteraceae, Aizoaceae and Poaceae. Forest, which also occupies a tiny area, supported 8.7% of species and a highly distinctive flora, including several tropical genera not present in other biomes (e.g. the grasses *Brachiaria* and *Dactylocnetum*, the trees *Apodytes* and *Clausena*, and the lianas *Cyphostemma* and *Dioscorea*). The flora of the Grassland biome (7.4%) was distinguished by the high numbers of Poaceae belonging to tropical ($C_4$) genera such as *Themeda*, *Cymbopogon* and *Eragrostis*. Disturbed areas such as roadsides, paths and sites cleared of dense alien vegetation, supported 4.2% of the flora that are typically "weedy" species (e.g., *Arctotheca calendula*, *Mesembryanthemum aitonis*, *Tribulus terrestris*).

Most species in our flora belong to the 62 genera endemic to the GCFR (24.4% of total) and 45 genera endemic to SA (15.2%) (Fig. 3B); thus almost 40% of the species in our flora are endemic to the SA Region, now afforded kingdom status (*Cox, 2001*). Tropical lineages comprised 24% of species, most of which were associated with 60 PanTr genera, for example, *Afrocanthium*, *Dovyalis*, *Olea* and *Zanthoxylum*. The 22 TrAfr genera include *Scadoxis*, *Searsia*, *Mystroxylon* and *Euclea*. A total of 20% of species were associated with temperate lineages, equally shared between those with distributions that are PanTe (16 genera, e.g., *Anchusa*, *Colchicum*, *Epilobium*, *Silene*) and TeAfr (34 genera, e.g., *Cyrtanthus*, *Helichrysum*, *Ficinia*, *Satyrium*). Cosmo genera such as *Carex*, *Juncus*, *Polygala* and *Senecio* comprised 16% of species.

About 47% of the Cape St Francis dune flora is endemic to the GCFR, most (40%) being CFR endemics (Fig. 3C). Species that extend into the Succulent Karoo biome of the GCFR do so along the Namaqualand coast, where they grow on coastal dunes, for example, *Dasispermum suffruticosum*, *Trachyandra divaricata* and *Salvia africana-lutea*. CFR endemics were mainly species restricted to vegetated dunes and coastal habitats (e.g., *Chasmanthe aethiopica*, *Restio eleocharis*, *Ficinia lateralis*) but also included many species with populations on dunes as well as the fynbos growing on the sandstones and shales of the hinterland, namely *Passerina corymbosa*, *Cullumia decurrens*, *Struthiola argentea*, *Osteospermum polygaloides* and *Muraltia squarrosa* (see Fig. 4 for examples). Tropical lineages in our flora have also produced species endemic to the coastal dunes of the CFR namely *Euclea racemosa* (which extends into the GCFR), *Olea exasperata*, *Robsonodendron maritimum* and *Rapanea gilliana*. Species endemic to the southeastern section of the CFR—so-called regional endemics—comprised nearly 15% of the flora; this group is almost entirely restricted to dune and coastal habitats, and includes many species associated with typical Cape clades, for example, *Capeochloa cincta* subsp. *sericea*, *Erica chloroloma*, *Felicia echinata* and *Phylica litoralis* (see Fig. 5 for examples). A smaller number, which are restricted to the coastal dunes of St Francis Bay and Algoa Bay (local endemics), included *Agathosma stenopetala*, *Aspalathus recurvispina*, *Othonna rufibarbis* and *Syncarpha sordescens* (see Fig. 5 for examples). Two regional endemics had tropical affinities, namely *Rapanea gilliana* and *Dovyalis rotundifolia*. Widespread species, comprising more than half the flora, included 34 dune specialists that extend into the subtropical east coast of South Africa and further afield, for example, *Gladiolus guenzii*, *Maytenus procumbens*, *Stipagrostis zeyheri* subsp. *barbata*, *Albuca nelsonii* and *Arctotheca populifolia* (see Fig. 6 for examples). Some of the widespread species are tropical species, for example, *Chionanthus foveolatus*, *Cassine peragua* and *Diospyros simii*. Others have Cosmo distributions, namely *Typha capensis* and *Phragmites australis*.

About 30% (120 species) of the St Francis flora is restricted to calcareous coastal substrata throughout their distribution (see Figs. 5 and 6 for examples). This included 31 species that also grow on calcarenite. No species was restricted to calcarenite. The 96 genera with coastal dune endemics included mostly Cape lineages, namely *Agathosma*, *Erica*, *Phylica*, *Restio* and *Carpobrotus*. Tropical lineages with calcareous dune endemics are *Cussonia*, *Euclea*, *Rapanea*, *Olea*, *Dovyalis*, *Maytenus*, *Robsonodendron* and *Searsia*.

## Biological traits

Woody growth forms comprised 41% of the flora (Fig. 7A). Dwarf shrubs (*Aspalathus*, *Indigofera*, *Delosperma*, *Thesium*) were most frequent (13.4%) among woody growth forms, and trees (e.g., *Afrocanthium*, *Chionanthus*, *Pittosporum*) were few, comprising 11 species (2.7%). Tall and low shrubs had roughly similar richness, the former largely comprising thicket species (*Olea*, *Euclea*, *Sideroxylon*, *Pterocelastrus*), the latter associated with fynbos lineages (*Agathosma*, *Erica*, *Phylica*, *Passerina*). The liana flora, associated with forest and thicket, was rich at 21 species (5.2%) of mostly tropical affinity (*Asparagus*, *Rhoicissus*, *Diospyros*, *Secamone*).

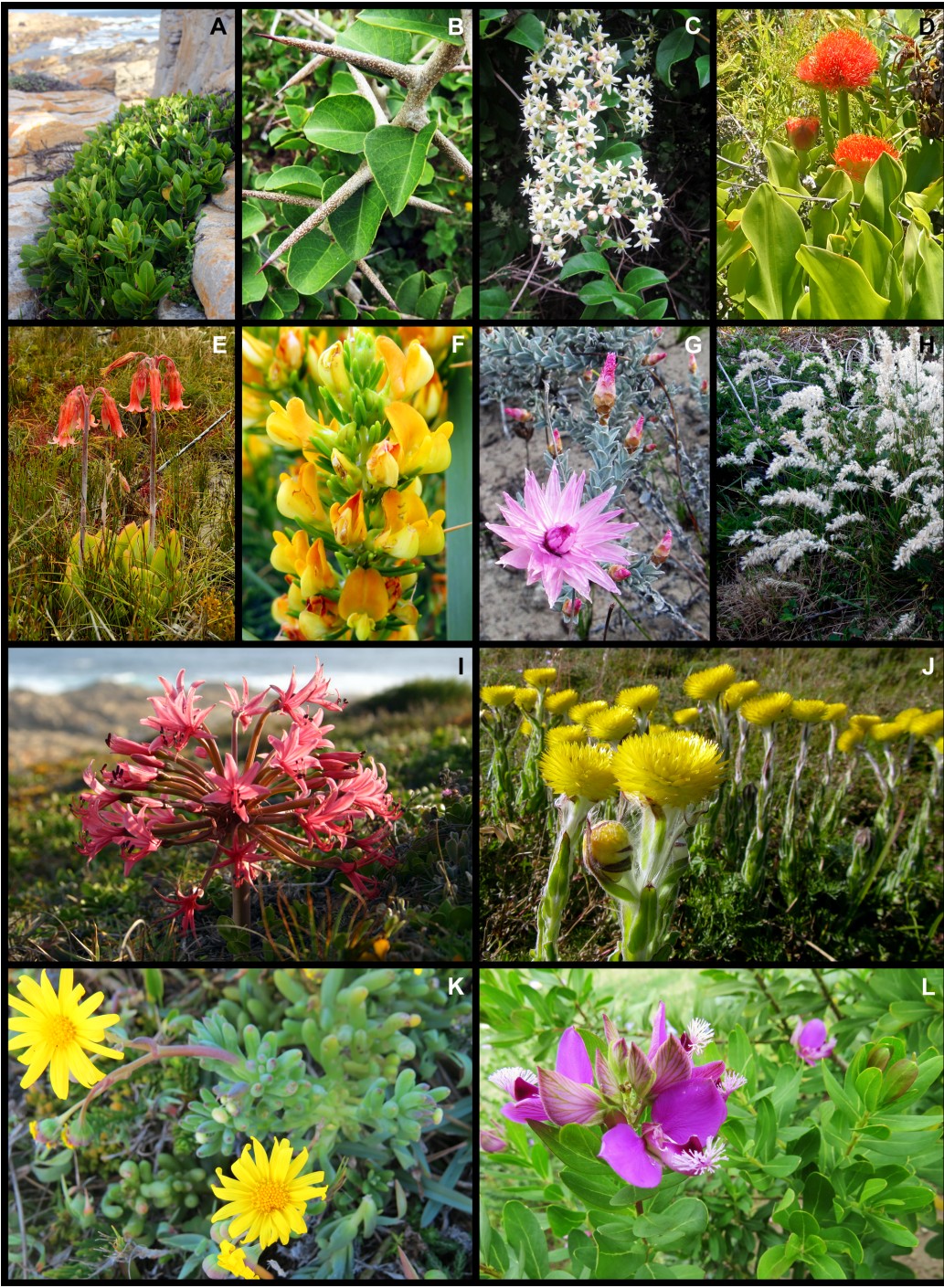

**Figure 4 Examples of plant species from the Holocene dune landscape around Cape St Francis in the southeastern Cape Floristic Region that occur on coastal dunes and on other substrata of the hinterland.** (A) *Sideroxylon inerme* subsp. *inerme*, typically a tall shrub in this landscape, here exhibiting an atypical wind-pruned form near the shore. (B) *Scolopia zeyheri*, (C) *Putterlickia pyracantha*, (D) *Scadoxus puniceus*, (E) *Cotyledon orbiculata*, (F) *Aspalathus spinosa* subsp. *spinosa*, (G) *Syncarpha canescens*, (H) *Melica racemosa*, (I) *Brunsvigia gregaria*, (J) *Helichrysum aureum* var. *aureum*, (K) *Crassothonna cacalioides*, (L) *Polygala myrtifolia* var. *myrtifolia*. Photos: Richard M. Cowling (A, C, D, G–J); Margie Middleton (B, F, K, L); B. Adriaan Grobler (E).

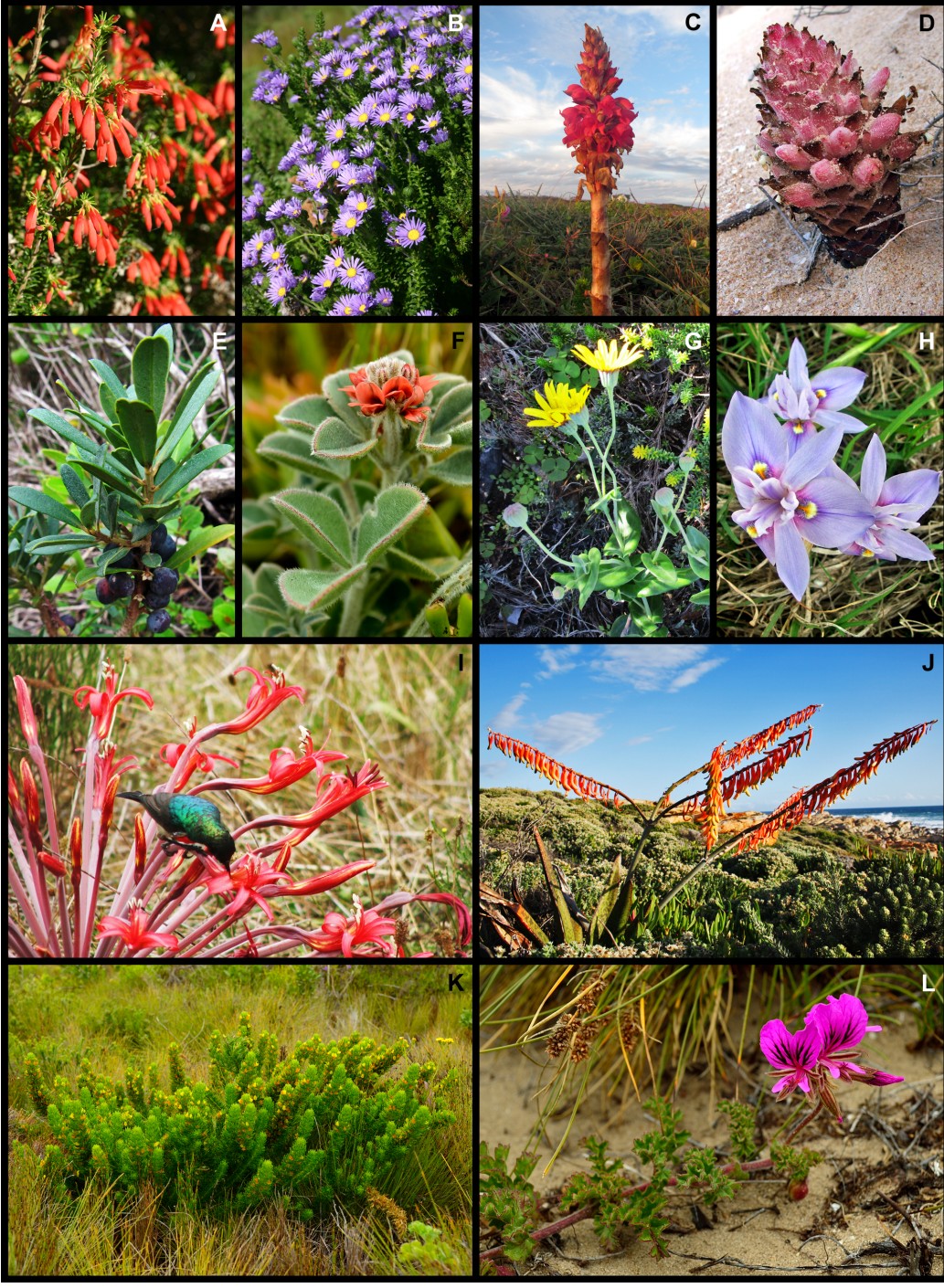

**Figure 5 Examples of regional and local dune endemic plant species from the Holocene dune landscape around Cape St Francis in the southeastern Cape Floristic Region.** (A) *Erica chloroloma*, (B) *Felicia echinata*, (C) *Satyrium princeps*, (D) *Hyobanche robusta*, (E) *Rapanea gilliana*, (F) *Indigofera tomentosa*, (G) *Othonna rufibarbis*, (H) *Moraea australis*, (I) *Brunsvigia litoralis* visited by a greater double-collared sunbird (*Cinnyris afer*), (J) *Gasteria acinacifolia*, (K) *Aspalathus recurvispina*, (L) *Pelargonium suburbanum* subsp. *suburbanum*. Photos: Richard M. Cowling (A–C, E, J); Caryl Logie (D, H, I); B. Adriaan Grobler (F, K, L); Margie Middleton (G).

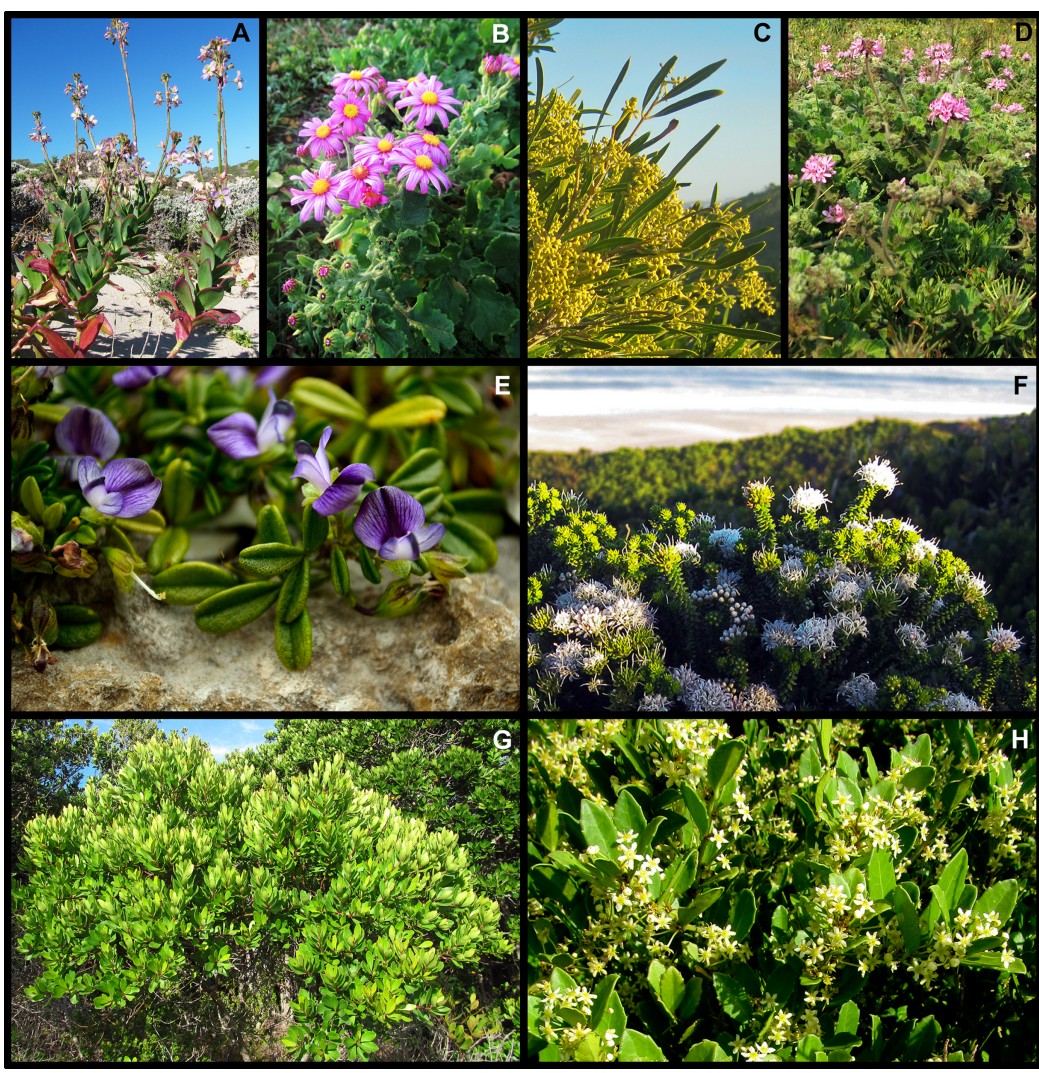

**Figure 6 Examples of widespread dune endemic plant species from the Holocene dune landscape around Cape St Francis in the southeastern Cape Floristic Region.** (A) *Heliophila linearis* var. *reticulata*, (B) *Senecio elegans*, (C) *Olea exasperata*, (D) *Pelargonium capitatum*, (E) *Psoralea repens*, (F) *Agathosma apiculata*, (G) *Euclea racemosa*, (H) *Maytenus procumbens*. Photos: Richard M. Cowling (A–C, F–H); Caryl Logie (D); B. Adriaan Grobler (E).

Among herbaceous growth forms, forbs (perennial herbs) were most common, comprising almost 20% of the flora (Fig. 7B). Species were concentrated in the Asteraceae, Orobanchaceae, Lobeliaceae and Scrophulariaceae. Hemicryptophytes comprised the second most frequent (17%) growth form, equally shared amongst species that are evergreen (Restionaceae, Cyperaceae) and deciduous (mainly Poaceae). Typical of GCFR floras, geophytes were diverse, with 55 species (14%), most of which were petaloid monocots (e.g., Iridaceae, Orchidaceae, Amaryllidaceae, Asphodelaceae). Vines, mostly thicket species (*Cynanchum, Clematis, Kedrostris, Dioscorea*) comprised 3.7% of the flora; thus, climbing plants (lianas and vines combined) approached 10% of species, a high tally for a warm-temperate flora.

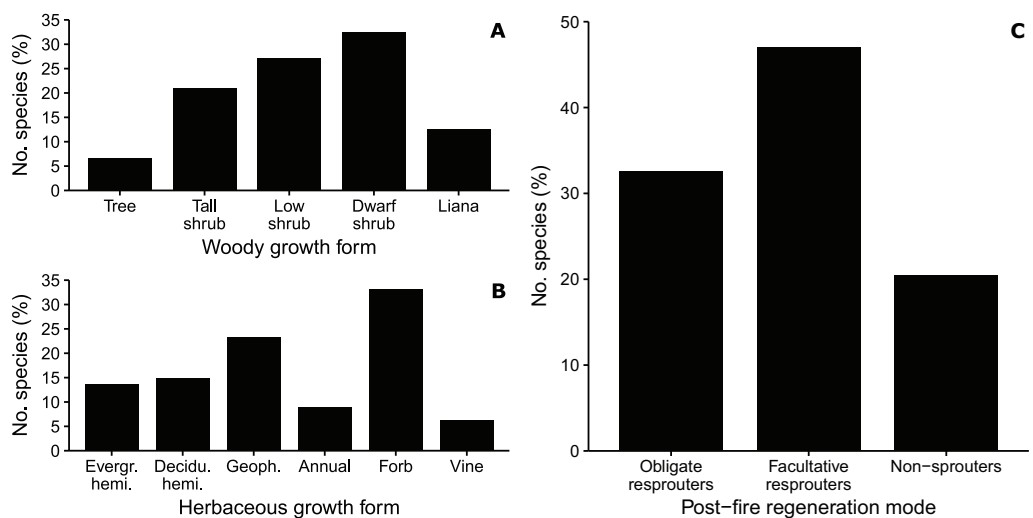

**Figure 7 Biological traits of a dune flora from Cape St Francis in the southeastern Cape Floristic Region.** (A) The proportion of species exhibiting different woody growth forms. (B) The proportion of species exhibiting different herbaceous growth forms; Evergr. hemi., evergreen hemicryptophyte; Decidu. hemi., deciduous hemicryptophyte; Geoph., Geophyte. (C) The proportion of woody species exhibiting different post-fire regeneration modes. See text for definition of growth forms.

Succulents comprised 20 species (5% of the total) in the St Francis flora, mostly Aizoaceae (*Delosperma, Carpobrotus*) but also Crassulaceae and Asphodelaceae.

Almost half (47%) of the woody species in the St Francis flora were categorized as facultative resprouters (Fig. 7C), regenerating post-fire from soil-stored seeds as well as sprouts from swollen root bases or lignotubers (none sprout from aerial shoots). All are components of fynbos and grassland, concentrated in the Asteraceae (*Helichrysum, Senecio*) and Fabaceae (*Aspalathus, Indigofera, Otholobium*). Obligate resprouters, which can resprout after fire but only establish seedlings in shaded microsites, comprised 33% of the flora. All obligate resprouters are forest and thicket shrubs and trees, with mostly bird-dispersed propagules, belonging to tropical lineages such as the Anacardiaceae, Asparagaceae, Celastraceae and Ebenaceae. Non-sprouters—species killed by fire— comprised 20% of the flora. Most of these belonged to typical fynbos genera (*Erica, Phylica, Agathosma, Muraltia*) and recruit seedlings only after fire; a small number, including species of *Helichrysum* and *Syncarpha*, recruit seedlings in between fires in physically disturbed habitats.

## Species of conservation concern

Nearly 8% (31 spp.) of the Cape St Francis dune flora comprise SCCs (Fig. 8; Table S1), most of which (26 spp.) are dune endemics. More than half (16 spp., 4%) of the SCCs are threatened: Vulnerable species (11 spp.) make up most of the threatened flora, but four Endangered (*Brunsvigia litoralis, Hyobanche robusta, Rapanea gilliana, Satyrium hallackii* subsp. *hallackii*) and one Critically Endangered (*Aspalathus recurvispina*) species also occur here. Most of the threatened species are either regional (six spp.) or local (six spp.)

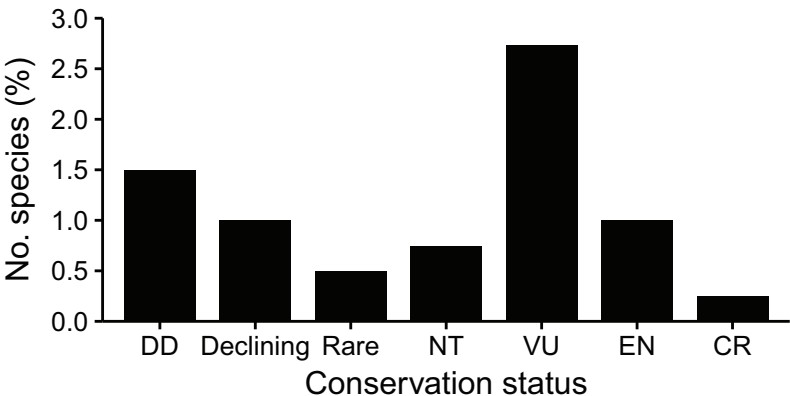

**Figure 8 Incidence of species of conservation concern (SCCs) in a dune flora from Cape St Francis in the southeastern Cape Floristic Region.** DD, data deficient; NT, near threatened; VU, vulnerable; EN, endangered; CR, critically endangered. See text for definitions.

dune endemics and were mainly associated with the Fynbos (eight spp.) and Coastal (four spp.) biomes. Additionally, two species (*Centella tridentata* var. *hermanniifolia*, *Delosperma saxicola*)—both local dune endemics—are listed as Rare. For all but one of the threatened species from Cape St Francis, coastal development and encroachment by alien invasive plants (especially *Acacia cyclops* and *Acacia saligna*) are the most severe threats; the exception, *Dioscorea sylvatica*, is primarily threatened by overharvesting for medicinal trade.

## DISCUSSION

### Flora composition

Families that are speciose in Cape fynbos landscapes such as Ericaceae, Restionaceae and Proteaceae (*Cowling & Holmes, 1992*) were poorly represented in the Cape St Francis flora as well as other coastal dune floras in the CFR (*Privett & Lutzeyer, 2010*; *Low, 2011*); indeed, Proteaceae are entirely lacking from these habitats, and ericas and restios have few representatives (*Cowling, 1983*).

Family composition on dunes is relatively homogeneous across the CFR although Iridaceae tend to increase in importance in the floras from the west of the CFR, at Walker Bay (*Privett & Lutzeyer, 2010*) and Table Bay (*Low, 2011*) (Table 1). Large genera in the Cape such as *Senecio, Helichrysum, Aspalathus, Ficinia* and *Pelargonium* are also relatively speciose on dunes. However, tropical lineages such as *Asparagus* and *Searsia* are well represented in Cape dune floras, a feature that distinguishes them from other floras in the CFR (*Cowling, 1983*).

Dune floras from the Mediterranean Basin are, like the St Francis flora, dominated by Fabaceae, Asteraceae and Poaceae (comprising together almost half the floras) with lesser contributions by the Amaranthaceae, Caryophyllaceae, Apiaceae and Plumbaginaceae (*Hadjichambis et al., 2004*; *Korakis & Gerasimidis, 2006*; *Ciccarelli, Di Bugno & Peruzzi, 2014*). These same three families are the most speciose in the dune floras of Mediterranean-climate California (*Purer, 1936*; *Barbour et al., 1981*; *US Fish & Wildlife Service, 2016*)

and Chile, which shows remarkable floristic affinity with the California flora (*San Martin, Ramírez & San Martin, 1992*). MCE dune environments in western and southern Australia are also dominated by Asteraceae, Fabaceae and Poaceae (*Opperman, 1999*; *Laliberté, Zemunik & Turner, 2014*); they share with the Cape and the Mediterranean Basin a high proportion of Cyperaceae and Amaranthaceae but are distinguished from all other MCE floras by having numerous species of Myrtaceae and several species of Proteaceae (*Dixon, 2011*).

## Geographical traits

There are few comparative data on biome representation in dune floras of the Cape and other MCEs of the world. At Table Bay, in the southwestern CFR, 66% of the dune flora is associated with Fynbos and 17% with Thicket (*Low, 2011*), proportions similar to those recorded for the St Francis flora. The Grassland and Forest biomes are not represented on the dunes of the Cape west coast (*Tinley, 1985*). Floras from other MCEs include a greater proportion than our flora of strictly coastal species (hummock dunes, rocky shores and cliffs), namely 17% in the Mediterranean Basin and 37% in California (*Barbour et al., 1981*). However, as in the Cape, most species are associated with fynbos analogs for these MCEs, namely dune matorral and chaparral. Dune forests in other MCEs are very species-poor, usually dominated by a small number of tree species belonging to *Quercus* and *Pinus* in the Northern Hemisphere (*Van Der Maarel & Van Der Maarel-Versluys, 1996*; *Peinado et al., 2007, 2011*) and *Eucalyptus* and other Myrtaceae in Australia (*Opperman, 1999*; *Dixon, 2011*).

Compared to the St Francis flora, genera endemic to the GCFR comprised a slightly higher proportion (30%) of the Table Bay dune flora, and a higher (26%) proportion of genera endemic to SA; thus, more than half the Table Bay species belong to genera endemic to SA (*Low, 2011*). Genera of tropical affinity were associated with 8% of species, a quarter of the number recorded in the St Francis flora. These data are consistent with the westwards decline in the diversity of species of tropical lineages in dune floras of the CFR (*Tinley, 1985*; *Cowling et al., 1997*).

There are no published analyses of the biogeography of supraspecific taxa in other MCEs. Genera of tropical origin are rare on coastal dunes of these MCEs. There are none in Western Australia (*Cowling et al., 1994*; *Dixon, 2011*), one in California (*Rhus integrifolia*) (*Barbour et al., 1981*), two in Chile (*Lithrea, Schinus*) (*Armesto, Arroyo & Hinojoa, 2007*), four in South Australia (*Alyxia, Logania, Bursera, Stackhousia*) (*Opperman, 1999*), and five in the Mediterranaean Basin (*Ceratonia, Pistacia, Phillyrea, Olea, Smilax*) (*Van Der Maarel & Van Der Maarel-Versluys, 1996*).

With regard to species-level endemism, the St Francis flora had a lower proportion than winter-rainfall Table Bay of GCFR endemics (75%) but similar proportion CFR endemics (38%) (*Low, 2011*). Most GCFR endemics in the Table Bay flora extend northwards into the semi-arid coast of Namaqualand (*Boucher & Le Roux, 1993*). Small dune fynbos floras derived from tenth-hectare plots sampled on the Agulhas Plain (500 km west of Cape St Francis) have 70–80% species endemic to the CFR (*Cowling & Holmes, 1992*), while for similar plots from the St Francis area, values are about 40% (*Cowling, 1983*),

similar to what we recorded at the flora level. Non-dune Cape floras, especially those compiled from sandstone fynbos, have much higher proportions of species endemic to the GCFR, approaching 100% in most cases (*Cowling, 1983*; *Cowling & Holmes, 1992*). Our flora is similar to those associated with grassy fynbos and renosterveld vegetation in the eastern CFR where GCFR endemics seldom comprise the majority of species (*Cowling, 1983*; *Kraaij, 2011*); it differs—as do other CFR dune floras (e.g., *Low, 2011*)—in that few non-GCFR taxa have tropical affinities.

There are few published data on the phytochorology and endemism of coastal dune floras of other MCEs. In the Mediterranean Basin, Mediterranean endemics comprise about 70% of regional floras (*Hadjichambis et al., 2004*; *Spanou et al., 2006*; *Muñoz Vallés, Gallego Fernández & Dellafiore, 2009*; *Ciccarelli, Di Bugno & Peruzzi, 2014*; *Iliadou et al., 2014*), except for Israel where a value of 45% was recorded (*Barbour et al., 1981*). Israeli coastal dunes support 26% of the species endemic to that country (*Kutiel, 2001*) and 21 species (6%) of the dune flora of Cyprus are endemic to the island (*Hadjichambis et al., 2004*). Overall, however, local endemism is highest in the western Mediterranean Basin (*Van Der Maarel & Van Der Maarel-Versluys, 1996*). A small (201 spp.) dune and coastal wetland from southern Spain included 25 species endemic to the Iberian Peninsula coast (*Muñoz Vallés, Gallego Fernández & Dellafiore, 2009*). Unlike the Cape, species with largely tropical distribution comprise only 1–2% of Mediterranean dune floras (*Barbour et al., 1981*; *Spanou et al., 2006*; *Muñoz Vallés, Gallego Fernández & Dellafiore, 2009*; *Iliadou et al., 2014*). Forty percent of California dune floras are MCE endemics and none have a tropical distribution (*Barbour et al., 1981*). "Narrowly endemic" species comprise 30% of a dune flora of Baja California (*Johnson, 1977*).

At 37%, a higher proportion of the Table Bay flora is endemic to calcareous substrata than at Cape St Francis (30%) (*Low, 2011*). Both floras lack species endemic to calcarenite, unlike dune floras of the southern CFR which includes on calcarenite outcrops species associated with the large flora endemic to cemented aeolianites (*Willis, Cowling & Lombard, 1996*).

High levels of edaphic endemism have been suggested for the dune floras of Chile (*Armesto, Arroyo & Hinojoa, 2007*) and the Mediterranean Basin (*Van Der Maarel & Van Der Maarel-Versluys, 1996*), but there are no data to illustrate this. About 18% of the floras of Israel's dunes are restricted to this habitat (*Barbour et al., 1981*; *Kutiel, 2001*); corresponding values are 37% for California (*Barbour et al., 1981*) and 40% for Baja California (*Johnson, 1977*).

## Biological traits

At Table Bay in the strongly winter-rainfall part of the CFR, woody plants comprise 39% of species (*Low, 2011*), a similar proportion as in the St Francis flora. However, trees are absent and lower stature woody plants (low and dwarf shrubs) dominate the woody flora (80%). The growth form mix of the herbaceous flora is similar to St Francis, except that annuals are more abundant (22 vs. 9%) and the incidence of forbs is lower (21 vs. 33%) (*Low, 2011*).

Growth form profiles of dune floras from the Mediterranean Basin differ in substantive ways to those from the CFR. Firstly, woody species contribute far less to the respective floras with figures ranging from 5% to 19%; trees are largely absent and dwarf shrubs predominate (*Barbour et al., 1981*; *Hadjichambis et al., 2004*; *Spanou et al., 2006*; *Ciccarelli, Di Bugno & Peruzzi, 2014*; *Croce et al., 2019*). Among herbaceous growth forms, annuals are by far the most speciose, comprising between 40% and 70% of floras; geophytes have similar proportions to the CFR (ca 10%) as do hemicryptophytes (ca 20%), but evergreen species are rare. In California, like the CFR dune floras, woody species comprise ca 40–50% of the total; all are shrubs and most of them dwarf in stature (*Barbour et al., 1981*; *Barbour, De Jong & Pavlik, 1985*). Both annuals and hemicryptophytes (mainly grasses) comprise 20–30% of California dune floras and geophyte incidence is low. The Chilean dune flora has a growth form mix similar to California, except for geophytes which are more frequent in Chile (*San Martin, Ramírez & San Martin, 1992*).

Most similar to the growth form mix of the St Francis flora, is the dune and calcarenite flora of southeastern South Australia (Fleurieu Peninsula to Port Macdonnell) (*Opperman, 1999*), a region physiographically similar to the Cape Agulhas area of the CFR. Features uniquely shared by these two floras are: multi-species tree floras; hemicryptophyte floras equally shared between deciduous and evergreen species; low (<10%) incidence of annuals; and high (ca 10%) incidence of lianas and vines (*Opperman, 1999*).

Native succulents are rare on the coastal dunes of other MCEs except for southern California where Cactaceae and Crassulaceae can make up 5–6% of floras (*Barbour et al., 1981*), comparable to the tally for Cape St Francis but half the value recorded at Table Bay (11%) (*Low, 2011*). Other than a few species of *Dudleya* in California, there is no equivalent in the other MCEs—in terms of growth form or diversity—of the dwarf, leaf-succulent shrubs in the Aiozaceae, that are so abundant and diverse in coastal dunes throughout the GCFR (*Boucher & Le Roux, 1993*; *Taylor & Boucher, 1993*).

A larger portion (28%) of the Table Bay woody dune flora comprise non-sprouters compared to the Cape St Francis flora (20%), whereas fewer (18%) are obligate resprouters (*Low, 2011*), in concert with the westwards decline in the diversity on CFR dunes of species associated with tropical thicket and forest lineages. The proportion of non-sprouters amongst the woody floras of mountain fynbos is much higher than on coastal dunes: of the 628 shrubs growing in sandstone fynbos of the Southern Langeberg mountains in the southern CFR, 80% are non-sprouters (*McDonald et al., 1995*).

There are no data on post-fire regeneration of component species of dune floras from other MCEs. We suspect, based on data on post-fire regeneration modes in other habitats, that the incidence of obligate resprouters and non-sprouters will be much lower than it is in the CFR (*Keeley et al., 2012*; *Rundel et al., 2018*): Western and South Australian dunes are likely to harbor non-sprouting shrubs in the Myrtaceae, Fabaceae and Ericaceae; in the Mediterranean Basin, some species of *Cistus* may be non-sprouters and in California, non-sprouters are likely to include dune species

of *Arctostaphylos* and *Ceanothus*; no non-sprouting shrubs are likely to be found on the coastal dunes of Chile.

## Phylogenetic structure

Dune floras of different regions are strongly phylogenetically dispersed, suggesting that these floras are mainly derived from adjacent, non-dune lineages and not by dispersal from geographically remote dunes (*Brunbjerg et al., 2014*). In the case of MCEs, it would appear that dune floras largely comprise species of recent (Plio-Pleistocene) origin (*Thompson, 2005*; *Hoffmann, Verboom & Cotterill, 2015*) that are derived from a subset of the adjacent, inland flora, mainly members of the Asteraceae, Fabaceae and Poaceae. In all cases for which data exists, most dune species are endemic to their respective MCEs and about 20–30% are endemic to calcareous dune sands; local endemism is generally high. While these floras have undergone diversification on colonizing dune habitats, there appears to have been limited radiation of lineages on dunes, either geographically or ecologically as evidenced by generally modest species-to-genus ratios (Table S2). Important exceptions include the radiation of *Limonium* on Mediterranean Basin dunes (*Van Der Maarel & Van Der Maarel-Versluys, 1996*).

Cape dune floras differ from other MCE dune floras in (1) the high numbers of species, especially dune endemics, associated with shrubby lineages, and (2) the relatively high incidence of species, including dune endemics, associated with tropical lineages. The first feature may simply be a consequence of the high plant diversity of the CFR (*Rundel et al., 2016*); there are many more lineages available to colonize coastal dunes than in other MCEs (diversity begets diversity). The second is a consequence of the unique geography of SA: nowhere else in the world does a MCE transition without any dispersal barriers to a subtropical region. Thus, the CFR has a permeable boundary with the tropics and coastal dunes have provided an effective corridor for tropical lineages to penetrate westwards, especially during mid to late Pleistocene glacials when summer rainfall conditions extended further west in the CFR than they do now (*Braun et al., 2019*; *Engelbrecht et al., in press*). This corridor has provided the basis for the unique phylogenetic structure of Cape dune floras.

## Convergence and divergence of growth form mix

The growth form mix of CFR dune floras showed strongest convergence with Australian dunes in that in both, evergreen hemicryptophytes and shrubs share dominance, and annuals are floristically subordinate. Furthermore, woody species—other than those of tropical affinity—are largely evergreen, leptophyllous and sclerophyllous (*Cowling et al., 1994*). In both regions, dune floras are assembled largely from older lineages that have diversified in nutrient-poor environments (*Hopper & Gioia, 2004*; *Hoffmann, Verboom & Cotterill, 2015*).

The least similar of MCEs to the CFR in terms of trait profile is the Mediterranean Basin. Here, annuals are the most frequent growth form while shrubs (mostly drought-deciduous dwarf species) are subordinate. In terms of growth form mix, California and Chile dune floras, which have a high component of evergreen shrubs and a

modest complement of annuals (*Barbour, De Jong & Pavlik, 1985*; *San Martin, Ramírez & San Martin, 1992*), occupy an intermediate position between the Cape and Australia on the one hand, and the Mediterranean Basin on the other.

## Flora conservation status

In the western CFR, SCCs comprise a similar proportion (7.5%) of the Table Bay dune flora compared to Cape St Francis, but the incidence of threatened species (2.4%) is lower (*Low, 2011*). Similarly, the Walker Bay flora has slightly lower incidences of SCCs (5.7%) and threatened species (3.4%) (*Privett & Lutzeyer, 2010*). Most threatened species from these western CFR dune floras are associated with the Fynbos biome and, as at Cape St Francis, coastal development and alien plant invasion are the primary threats (*SANBI (South African National Biodiversity Institute), 2017*).

Data on SCCs from dune landscapes in other MCEs are sparse, existing mainly for floras from the Mediterranean Basin where the incidence of SCCs and threatened species is comparable to that in the Cape St Francis flora: in Cyprus, *Hadjichambis et al. (2004)* noted 18 "Red Data Book" species occurring in dunes, constituting about 5% of the Cypriot sand dune flora; a small (11.8 ha) remnant of dune vegetation on the southern coast of Spain contains four "threatened and vulnerable" species (2.8%) in its flora (*Gómez-Zotano, Olmedo-Cobo & Arias-García, 2017*); the coastal dune flora of northern Campania in Italy comprises 10 SCCs (3.6%), but most of these are regional endemics or rarities and only one species is classified as Vulnerable (*Croce et al., 2019*). In the Guadalupe-Nipomo dunes of Mediterranean-climate California, 19 species (13.5%) are considered "endangered, threatened or rare," although only five (3.5%) of these are listed as Endangered or Threatened at the state or federal level (*US Fish & Wildlife Service, 2016*). The coastal dunes of South Australia host 36 species that are "endangered, vulnerable or rare"; for the limited sites where data are available, these typically constitute 1–2% of local dune floras (*Opperman, 1999*). While the proportion of SCCs in MCE dunes vary from place to place, it is evident that coastal dune floras consistently comprise species that are vulnerable to extinction.

Dune ecosystems are under threat everywhere, from alien plant invasions, ongoing conversion for tourism, urban and industrial use to increase in sea-level as a result of global warming (*Brown & McLachlan, 2002*; *Defeo et al., 2009*; *Prisco, Carboni & Acosta, 2013*). In the CFR, most Holocene dune vegetation types have a low-priority conservation status ("Least Threatened"), based on the extent to which conservation targets have been, or still have the potential to be achieved (*Rouget et al., 2014*). However, conservation assessments need to consider the rarity of the coastal dune habitat (*Van Der Maarel & Van Der Maarel-Versluys, 1996*): endemics are restricted to small, dynamic and fragmented habitats relative to more extensive zonal ecosystems. Cape endemics may well be members of communities "super-saturated" with species recruited from those that occupied the extensive glacial dunefields of the vast expanse of the Palaeo-Agulhas Plain (PAP) that was exposed at lower sea levels (HC Cawthra, RM Cowling, S Andò, CW Marean, 2019, unpublished data; BA Grobler, HC Cawthra, AJ Potts, RM Cowling, 2019, unpublished data); many range-restricted and rare species

may be vulnerable to extinction owing to small population size effects. This is likely to be the case for other MCEs. We propose that all Holocene dune vegetation in all MCEs should be red listed as should all dune endemics.

## CONCLUSION

The Pleistocene physiographic dynamics of the Cape south coast are relatively well understood, typified by the rapid sea level changes that exposed the PAP which, at maximum extent, approximated the size of the extant CFR (CW Marean, RM Cowling, J. Franklin, 2019, unpublished data). The PAP was a dynamic landscape and young calcareous dunes covered vast areas, especially when compared to the tiny and fragmented areas that Holocene dunes occupy today (HC Cawthra, RM Cowling, S Andò, CW Marean, 2019, unpublished data). The coastal dune systems of the CFR offer numerous opportunities to research aspects of the most recent diversification event in the CFR—the evolution of a flora endemic to this habitat (*Hoffmann, Verboom & Cotterill, 2015*). Thus, a major research question is: how did these dynamics influence the phytogeography, evolution, and diversity of the coastal dune flora? While our focus is on terrestrial vascular plants, the approach we outline here is equally applicable to other groups of organisms. Ultimately, this study is about highlighting an outstanding model system for explicitly testing plant evolutionary questions. Important questions that need to be addressed include:

1. What is the size, composition, geographical, biological, and ecological traits of extant flora associated with Holocene coastal dunes on the CFR coast?
2. What are the geographical patterns of calcareous dune endemism and are they related to offshore (historical) or onshore (contemporary) physiographic and vegetation features?
3. What are the estimated times of diversification of endemic and non-endemic clades and do endemic clades show differences in biological traits, i.e., a coastal adaptation?
4. Do populations of non-endemic species that grow on calcareous coastal substrata show differences in biological traits from populations that grow on inland, acid substrata, and do these differences parallel those between calcareous-endemic and non-endemic sister species?
5. Can phylogeographic analysis of disjunct endemics provide insights on the timing of population fragmentation in relation to Pleistocene sea level fluctuations?
6. Can offshore habitat composition and extent predict any of the diversification patterns and processes posited above?
7. What are the implications of these biodiversity dynamics for human evolution and persistence on the Cape south coast?

## ACKNOWLEDGEMENTS

We thank Jal and Jan Rigaard from Rocky Coast Farm for permission to access the property. Thank you also to the iNaturalist community for assistance with the identification of plant species.

### Funding

This work was funded by the National Research Foundation and Foundational Biodiversity Information Programme (Integrated Biodiversity Small Grant UID 110438). B. Adriaan Grobler is supported by a National Research Foundation postdoctoral fellowship (Grant No. 116756). The funders had no role in study design, data collection and analysis, decision to publish, or preparation of the manuscript.

### Grant Disclosures

The following grant information was disclosed by the authors:
National Research Foundation and Foundational Biodiversity Information Programme (Integrated Biodiversity Small Grant UID 110438).
National Research Foundation postdoctoral: 116756.

### Competing Interests

Richard M. Cowling is an Academic Editor for PeerJ.

### Author Contributions

- Richard M. Cowling conceived and designed the experiments, performed the experiments, analyzed the data, contributed reagents/materials/analysis tools, prepared figures and/or tables, authored or reviewed drafts of the paper, approved the final draft.
- Caryl Logie performed the experiments, contributed reagents/materials/analysis tools.
- Joan Brady performed the experiments, contributed reagents/materials/analysis ools.
- Margie Middleton performed the experiments, contributed reagents/materials/analysis tools.
- B. Adriaan Grobler performed the experiments, analyzed the data, contributed reagents/materials/analysis tools, prepared figures and/or tables, authored or reviewed drafts of the paper, approved the final draft.

### Field Study Permissions

The following information was supplied relating to field study approvals (i.e., approving body and any reference numbers):

Collection of plant specimens was approved by the Provincial Administration of the Eastern Cape Province: Chief Directorate Environmental Affairs (permit to pluck flora No. CRO 138/18CR).

### Data Availability

The full list of plant species from the Cape St Francis Holocene dune landscape, together with their traits, is available in the Supplemental File.

## Supplemental Information

Supplemental information for this article can be found online at http://dx.doi.org/10.7717/peerj.7336#supplemental-information.

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
