# Peer review of "Taxonomic, biological and geographical traits of species in a coastal dune flora in the southeastern Cape Floristic Region: regional and global comparisons"

_PeerJ, doi:10.7717/peerj.7336_

## Round 0.1 · original submission · Major Revisions

Once I have received the review reports, I feel that your manuscript is interesting, and potentially publishable as a valuable contribution towards the comparative knowledge of Mediterranean climate ecosystems, but that major revisions have to be performed in order to be acceptable for publication in PeerJ.

You will see that one of the reviewers (reviewer 2) makes very specific recommendations for a thorough revision of the manuscript (see the attached review report). Please follow the points addressed by this reviewer when carrying your revision. A special emphasis should be made to improve the focus and structure of the manuscript, as suggested by this reviewer, with whom I fully agree.

Please carefully consider also point 4 addressed by reviewer 1 regarding the clarification of the criteria used for the assignment of species to the different biogeographical units of the Cape Floristic Region.

When submitting your review, please include a rebuttal letter in which you should comment how the different concerns raised by the reviewers were addressed.

·

Basic reporting

The manuscript is written in very good English with a few typographical, grammatical and general errors which I have indicated below.
Literature references and sufficient field background and context are provided.
The article conforms to the structure required by PeerJ.
Relevant results are presented on a documented flora. As this is descriptive research, this research is not strictly hypothesis driven.

Experimental design

The experimental design of this study is simple although labor intensive. It involves the careful documentation and identification of the flora of the area and analysis according to geographical traits, biological traits and species conservation status.

Validity of the findings

In this study, the coastal dune flora of the Cape St Frances area has been assembled over a number of years by experts in the field, and identifications were made by them and by using iNaturalist. This data is summarized in appropriate bar graphs and tables as well as a species listing of the flora in a supplementary file. Consequently, the study can be viewed to be original, rigorous, and the findings valid. Some clarification is required for the presentation of some of the data as I have indicated in the comments to the author.

Additional comments

I wish to bring the following points to the attention of the authors:
1. The reference, Le Roux, (2000), as referred to in Line 130 is not in the reference list.
2. Line 92 and 93: At this stage, some 13 900 km2 of the PAP is comprised of calcareous dunes, a 24-fold increase relative to the present time. This last phrase does not make sense in relation to the previous sentence and needs rephrasing. Should this not be a decrease?
3. Line 130: The sentence “Contrary to Le Roux (2000), no Cenozoic aeolianites (Nanaga Formation) occur in the region.” needs clarification. This is too brief to be understandable as it gives the reader no indication of the area to which it is compared to, and needs a phrase such as “Contrary to Le Roux (2000), who investigated the …… of ….., no Cenozoic aeolianites (Nanaga Formation) occur in the region.”
4. The authors make use of the area definitions (defined in lines 226 to 230) of the Greater Cape Floristic Region (GCFR) as defined by Born et al. (2007), the Cape Floristic Region (CFR) as defined by Manning and Goldblatt (2012) and the southeastern centre of the CFR (CFRse). A reference for the definition of the CFRe should be given. Species are then classified according to these criteria, and the results of these analyses are shown in Figure 3 C. According to the classification, the CFRe forms a smaller part of the area of the CFR, whilst the CFR in turn forms part of the even larger GCFR. Thus, in my opinion, if a species is scored as present in the CFRe, it can also be scored as present in the CFR and in turn in the GCFR. If one inspects the bar graph in Figure 3 C, this is not apparent, because then the GCFR bar should be the tallest representing the cumulative number of these species. This needs clarification please. I assume that species that have been scored for presence in the CFRe, have not been scored as present in the rest of the CFR and the rest of the GCFR, and the species occurring in the CFR have not been scored in the GCFR. This needs clarification in the methods or else this is not understandable, and the statistics do not make sense.
5. It s not clear from the presentation of the data, who generated the floras of Walker Bay and Koeberg. References to the authors of these floras should be specifically indicated in the text in lines 286-288 and again in Table 1, as this data was not generated as part of this study.
Typographical and grammatical corrections are required as follows, and are indicated:
Line 66: “ coastal dunes, especially on their seawards margin, are subject to frequent, strong, salt-laden winds…”
Lines 133 to 135: due to the long sentence construction the end of the sentence is not correctly structured (plural required of "systems", as indicated): “Geomorphologically, the study area comprises vegetated parabolic dunes and several more recent plumes of mobile sand forming and impressive (maximum 14 km-long) bypass dune systems (Tinley, 1985; Illenberger and Burkinshaw, 2008).
Line 135: Add comma for clarity: On the inland margin, dunes reach a maximum altitude of 62 m.
Line 166: spelling Salvia africana-lutea
Line 183: spelling Senecio halimifolius,
Line 216: spelling Pterocelastrus tricuspidatus
Line 360: There “are” none on California (not “is”)..
Line 388: Fig. 6 Some of the wide-spread species (not “wides”)
Line 419: are MCE endemics and none have a tropical distribution….
Line 460: “Trees, however, trees are absent and lower……” delete “Trees”and start sentence with “However”.
Line 517: “Fababceae” should be spelt “Fabaceae”
Line 520 and 521: “…..no non-sprouting shrubs are unlikely to be found on the coastal dunes of Chile.” The double negative needs rephrasing.
Line 589: Two errors: “The growth form mix of CFR dunes floras shows strongest convergence with Australian dunes in that in both, evergreen hemicryptophytes and shrubs share dominance, and annuals are floristically and subordinate.” The first "dunes" should not be plural and delete the last “and”.

·

Basic reporting

I think the paper is too long with too much of the data presented in the text, and I suggest using tables or figures for much of this data.

I also think the paper would benefit from some restructuring and a tighter focus - see attached review.

Experimental design

No comment.

Validity of the findings

The findings are valid and interesting.

Additional comments

Please see attached review file.

---

## Round 0.2 · accepted · Accept

Thank you very much for the submission of a revised version of your paper. I have gone through the revised, track-changes manuscript and rebutal letter and see that the concerns raised by the reviewers have been thoroughly addressed. As a result, I think that the paper has been considerably improved and may be now accepted for publication in PeerJ. Thank you once again for your dedicated work on the revision.

The editorial office have notwithstanding informed me that it is PeerJ´s standard policy to require a brief Conclusion section for every submission. I know that you have specifically asked for removing a conclusion section, but I think you may extract a few relevant points from the abstract or the paper itself. A short round-up paragraph or key findings (not necessarily repeating the abstract which is already quite long) would be sufficient.

Congratulations and best wishes,

Santiago